# UV-Vis Activated Cu_2_O/SnO_2_/WO_3_ Heterostructure for Photocatalytic Removal of Pesticides

**DOI:** 10.3390/nano12152648

**Published:** 2022-08-01

**Authors:** Alexandru Enesca, Luminita Andronic

**Affiliations:** Product Design, Mechatronics and Environmental Department, Transilvania University of Brasov, Eroilor 29 Street, 35000 Brasov, Romania; andronic-luminita@unitbv.ro

**Keywords:** pesticides, wastewater, photocatalysis, metal oxides, semiconductors

## Abstract

A three-steps sol–gel method was used to obtain a Cu_2_O/SnO_2_/WO_3_ heterostructure powder, deposited as film by spray pyrolysis. The porous morphology of the final heterostructure was constructed starting with fiber-like WO_3_ acting as substrate for SnO_2_ development. The SnO_2_/WO_3_ sample provide nucleation and grew sites for Cu_2_O formation. Diffraction evaluation indicated that all samples contained crystalline structures with crystallite size varying from 42.4 Å (Cu_2_O) to 81.8 Å (WO_3_). Elemental analysis confirmed that the samples were homogeneous in composition and had an oxygen excess due to the annealing treatments. Photocatalytic properties were tested in the presence of three pesticides—pirimicarb, S-metolachlor (S-MCh), and metalaxyl (MET)—chosen based on their resilience and toxicity. The photocatalytic activity of the Cu_2_O/SnO_2_/WO_3_ heterostructure was compared with WO_3_, SnO_2_, Cu_2_O, Cu_2_O/SnO_2_, Cu_2_O/WO_3_, and SnO_2_/WO_3_ samples. The results indicated that the three-component heterostructure had the highest photocatalytic efficiency toward all pesticides. The highest photocatalytic efficiency was obtained toward S-MCh (86%) using a Cu_2_O/SnO_2_/WO_3_ sample and the lowest correspond to MET (8.2%) removal using a Cu_2_O monocomponent sample. TOC analysis indicated that not all the removal efficiency could be attributed to mineralization, and by-product formation is possible. Cu_2_O/SnO_2_/WO_3_ is able to induce 81.3% mineralization of S-MCh, while Cu_2_O exhibited 5.7% mineralization of S-MCh. The three-run cyclic tests showed that Cu_2_O/SnO_2_/WO_3_, WO_3_, and SnO_2_/WO_3_ exhibited good photocatalytic stability without requiring additional procedures. The photocatalytic mechanism corresponds to a Z-scheme charge transfer based on a three-component structure, where Cu_2_O exhibits reduction potential responsible for O_2_ production and WO_3_ has oxidation potential responsible for HO· generation.

## 1. Introduction

The agricultural sector is considered one of the larger water consumers. However, due to population growth and climate change, the water shortage represents an important challenge to overcome [1,2]. The recycled water used in crop irrigation usually contains high quantities of pesticides, due to inefficient wastewater-treatment plant technologies. The risk is even higher when the pesticides interfere with the food chain as a result of transfer from water to soil and from soil to plants through roots [3,4,5].

Pesticide consumption in agricultural areas has significantly increased in the last few years. Only in 2014 was the pesticide consumption reported by FAOSTAT (North America was not included) above 3,000,000 tons [6]. Epidemiological studies have shown the carcinogenic potential of certain pesticides able to produce birth defects, endocrine deficiencies, cardiovascular diseases, or even disorders of the reproductive organs [7,8,9]. Traditional methods used to reduce pesticides form wastewater, such as ozonation or chloride dioxide insertion, are effective ways to address the problem, but raise economic and environmental issues [10,11]. Chlorine dioxide (ClO_2_) is considered an alternative to NaClO products able to work as a disinfection agent for the wastewater reused in the agricultural sector. However, the ClO_2_^−^ and ClO_3_^−^ biproducts formed during the water treatment can damage the thyroid gland and cause anemia [12,13].

Photocatalysis stands out as a modern and sustainable technology based on advanced oxidation processes (AOPs). In terms of function of the photocatalytic materials involved in the process, the wavelength may vary from UVA to the entire visible spectrum [14,15]. Under irradiation, the catalysts generate oxidative and superoxidative species that promote a complex series of oxidation reactions, aiming to induce the complete mineralization of the organic pollutants [16]. Monocomponent catalysts, such as TiO_2_ [17], CuO [18], SnO_2_ [19], WO_3_ [20], ZnO [21], and Cu_2_S [22], exhibit a series of disadvantages, such as limited light absorption, low chemical stability, and fast charge carrier recombination. In order to overcome these issues, heterostructures and composite materials have been developed. Similar systems based on metal oxide heterostructures like TiO_2_/WO_3_/ZnO [23], BiOI/BiOCl [24], BaTiO_3_/KNbO [25], TiO_2_/CuO [26], and CuO/Bi_2_WO_6_ [27] exhibit lower charge recombination rates and high photocatalytic efficiency toward dye molecules. The presence of sulfur compounds as part of heterostructure matrices, such as SnS_2_/ZnInS_4_ [28], CoTiO_3_/ZnCdS_2_ [29], CdS/CeO_2_ [30], CoWO_4_/CoS [31], and ZnCdS_2_/MoS [32], extend the light absorbance spectrum without solving the chemical stability issues in acid or alkaline environments. Composite materials based on g-C_3_N_4_ [33], g-C_4_N_6_ [34], and r-GO [35] exhibit high surface-active materials able to efficiently eliminate the pollutants using both absorption and photocatalytic processes.

In this work, the photocatalytic activity of Cu_2_O/SnO_2_/WO_3_ was evaluated toward three pesticides—pirimicarb (PIR; insecticide), S-metolachlor (S-MCh; herbicide) and metalaxyl (MET; fungicide)—used in the agricultural sector. The pollutants were selected based on their toxicity and long-term persistence in soil and water. Pesticide diffusion into plants is an important source of food contamination that then expands in the human body with negative impacts on long-term health. A heterostructure powder was obtained by a three-step sol–gel synthesis followed by spray deposition as film. Each component was chosen based on its spectral absorbance and suitable position of the energy bands, able to reduce the charge carrier’s recombination rate and improve long-term photocatalytic activity. The synergic action of high Cu_2_O visible absorption and facile charge migration through the heterostructure provides a good alternative for oxidative species generation required for pollutant mineralization. Additionally, the study focused on free-titania photocatalysts as an alternative to traditional materials. Crystalline composition influence on surface composition and film morphology were studied. Photocatalytic efficiency considers the mechanism of oxidative species generation and includes UV-vis and TOC/TN evaluation. The results indicate that the Cu_2_O/SnO_2_/WO_3_ heterostructure is suitable for pesticide removal and can be considered a sustainable future alternative.

## 2. Experimental

### 2.1. Material Synthesis and Heterostructure Deposition

#### 2.1.1. Heterostructure Powder Synthesis

The heterostructure synthesis consisted of a three-step sol–gel (Figure 1) method followed by thermal treatments to remove all the solvent content.

Step 1. WO_3_ powder was obtained from a precursor composed of tungsten hexachloride (99.8%, WCl_6_, AcrosOrganics, Gell, Germany) dissolved in a mixed solvent of absolute ethanol (100%, C_2_H_6_O, Sigma Aldrich, Munich, Germany) and 2-propanol (100%, C_3_H_8_O, Sigma Aldrich, Munich, Germany). The precursor was magnetically stirred at 140 rpm for 240 min in the dark until a yellow homogeneous solution formed. Gel development took place after the drop-by-drop addition of 0.24 mol sodium hydroxide (99.98%, NaOH, Honeywell, Charlotte, NC, USA). The gel was kept in the dark for 12 h and the final precipitate was centrifuged. The yellow powder (Figure 1, step1) was thermally treated at 410 °C for 4 h.

Step 2. SnO_2_/WO_3_ powder was prepared by inserting the WO_3_ powder previously obtained into a precursor based on tin tetrachloride (99.7%, SnCl_4_, Sigma Aldrich, Munich, Germany) dissolved in absolute methanol (100%, CH_4_O, Sigma Aldrich, Munich, Germany). The mixture was stirred at 140 rpm for 220 min in dark to ensure the uniform dispersion of WO_3_ powder into the SnO_2_ precursor. By slowly adding 0.18 mol sodium hydroxide (99.98%, NaOH, Honeywell, Charlotte, NC, USA) a yellowish white precipitate (Figure 1, step 2) formed, which was centrifuged and annealed at 410 °C for 6 h.

Step 3. the Cu_2_O/SnO_2_/WO_3_ heterostructure was developed by adding the SnO_2_/WO_3_ powder previously obtained into a precursor composed of 0.03 mol copper acetate (Cu(CH_3_COO)_2_, 97%, Sigma Aldrich, Munich, Germany) and deionized water. After 120 min of magnetic stirring at 140 rpm in dark, a mixture of NaOH, glucose (C_6_H_12_O_6_, 99.8%, Sigma Aldrich, Munich, Germany), and deionized water was slowly added until the precipitate had formed. Due to the copper tendency to change from the +1 to +2 oxidation state, after centrifugation the reddish powder (Figure 1, step 3) was dried for 12 h at room temperature.

#### 2.1.2. Film Deposition

The heterostructure films (Figure 1, step 4) were obtained by cold spray deposition. The deposition precursor was obtained by dispersing 50 mg Cu_2_O/SnO_2_/WO_3_ powder in a 50 mL mixture of ethanol and 2-propanol by ultrasound bath. A small quantity (0.1 mL) of Triton X was added to the precursor and magnetically mixed for 30 min. The 2 × 2 cm^2^ microscope glass substrate was firstly degreased and then cleaned by successive immersion in acetone and ethanol. The clean substrate was preheated at 45 °C for 120 min and then the precursor was sprayed at 0.5 bars. Breaks of 12 min were kept between each deposition sequence in order to allow solvent evaporation.

### 2.2. Photocatalytic Experiments

The photocatalytic experiments were done using a rectangular photoreactor containing 4 UVA light sources (18 W, black tubes, T8, 320–370 nm, λ_max_ = 360 nm, 3Lx flux intensity, Philips) and 4 Vis sources (18 W, cold tubes, TL-D 80/865, 400–700 nm, λ_max_ = 560 nm, 28 Lx flux intensity), a ventilation engine to ensure stable humidity, and temperature sensors required to keep a stable 25 °C during the experiments. The light sources were placed in radial position to provide maximum light radiation and ensure uniform light distribution. The energy of the incident light on the sample was 12.63 mW/cm^2^.

Pirimicarb (PIR; insecticide, C_11_H_18_N_4_O_2_, 98%, Sigma Aldrich, Munich, Germany), S-metolachlor (S-MCh; herbicide, C_15_H_22_ClNO_2_, 98.5%, LGC, Augsburg, Germany), and metalaxyl (MET; fungicide, C_15_H_21_NO_4_, 98%, Sigma Aldrich, Munich, Germany) were used to prepare 50 mg/L aqueous solution. In 40 mL pesticide solution was inserted one 2 × 2 cm^2^ piece of microscopic glass covered with Cu_2_O/SnO_2_/WO_3_. Quartz recipients were used during the experiments. The samples were kept in the dark for 120 min to reach the k5 absorption–desorption equilibrium. Light exposure duration was 10 h and photocatalytic evaluation was done hourly.

### 2.3. Characterization

The crystalline composition was evaluated with an X-ray diffractometer (model D8 Discover, Bruker, Karlsruhe, Germany) using the locked-couple option. The scan step was 0.002 degree with a rate of 0.020 s/step from a 20 to 60 degree theta angle. The samples’ morphology was studied in a high-vacuum regime with scanning electron microscopy (SEM; S–3400 N type 121 II, Hitachi, Tokyo, Japan) at an accelerated voltage of 10 kV. Field-emission scanning electron microscopy (FESEM SU8010, Fukuoka, Japan) was also used for morphological characterization, where the samples were covered with gold coating due to the low surface conductivity. The changes in the specific absorbance for each pollutant were registered by UV-vis spectrometry (Lambda 950, PerkinElmer, Waltham, MA, USA). The optical band gap energy of the single-component samples was estimated by the Wood and Tauc model. Pollutant mineralization was measured using a total organic carbon analyzer (TOC-L, model CPN, Shimadzu, Kyoto, Japan) and a total nitrogen analyzer (TNM-L, model RHOS, Shimadzu, Kyoto, Japan). The system was set to make three consecutive injections and to provide a mediated value. This evaluation provides essential information regarding the possible formation of by-products that are not identified during the UV-vis analysis.

The photocatalytic decolorization was evaluated by the UV-vis method. The changes in pollutant concentration were measured by UV-vis spectrometry based on the calibration curve corresponding to each pollutant. The absorption wavelength characteristics for each pollutant were PIR 315 nm, S-MCh 274 nm, and MET 267 nm. The photocatalytic efficiency was calculated based on the initial (C_0_) and the final concentrations (C) using the following equation:(1)$$\eta =\left[\frac{\left(C_{0}-C\right)}{C_{0}}\right]\cdot 100$$

## 3. Results and Discussion

### 3.1. Composition and Morphology

The crystalline composition of each sample was evaluated by X-ray diffraction, and the results are presented in Figure 2. All the analyses were done on films obtained from the same quantity of precursor without additional thermal treatment. The monocomponent sample contained WO_3_ with a monoclinic structure (ICCD 83-0951). The bicomponent sample contained monoclinic WO_3_ and tetragonal SnO_2_ (ICCD 41-1445). After the last synthesis step, the heterostructure exhibits all three metal oxides components: monoclinic WO_3_, tetragonal SnO_2_, and cubic Cu_2_O (ICCD 71-3645). The synthesis procedure allows WO_3_ and SnO_2_ to act as nucleation sites for the following component, which facilitates the formation of crystalline structure. There was no indication of mixed metal oxide formation, but the presence of amorphous compounds cannot be excluded [36,37]. Due to the synthesis procedure, no preferential orientation was observed. In order to avoid the formation of mixed copper oxides, the heterostructure powder was dried at room temperature for 12 h and the films deposited at 45 °C. Without annealing treatment, the persistence of carbonaceous species originating from the sol–gel synthesis is possible and may influence the photocatalytic activity.

The changes in crystallite sizes were calculated with the Scherrer formula in Equation (2), using the previous diffraction analysis [38]:(2)$$D=\frac{0.9\lambda }{\beta \cos\theta }$$
where λ is the X-ray wavelength value (1.5406 Å for CuK_α1_), β is the angular width measured at half-maximum intensity (FWHM) of the most representative peak, and θ is represents the Bragg’s angle. The instrumental broadening was considered during the calculations. The crystallite sizes are included in Table 1 for the WO_3_ and SnO_2_ components annealed at the same temperature, but for different periods. Using WO_3_ as a substrate for SnO_2_ development will have a negligible impact on the crystalline size of the first component. Both metal oxides exhibit similar crystallite size, which may be related to the annealing process taking place at the same temperature. The annealing period was higher for WO_3_, which shows the highest crystallite sizes. The same results are obtained when SnO_2_/WO_3_ is used as substrate for Cu_2_O development. However, the absence of annealing on the last synthesis step will induce the formation of lower crystallite size values corresponding to the Cu_2_O component. This behavior was also reported in other works [39,40], showing that if higher temperature is used the copper will pass in both oxidation states.

Sample morphology was evaluated by SEM analysis, and the results are presented in Figure 3. The monocomponent sample exhibits fiber-like morphology (Figure 3a) corresponding to WO_3_, which serves as substrate for SnO_2_ development. The fibers have a random distribution in terms of diameter and length. The addition of the second component (SnO_2_) will completely cover the WO_3_ fibers and the morphology resembles overlapping pallets (Figure 3b). The final heterostructure containing all three components presents a porous morphology (Figure 3c), with Cu_2_O uniformly distributed on the sample surface. Due to the low surface conductivity and the presence of aggregates, it was difficult to properly evaluate the grain size corresponding to the Cu_2_O/SnO_2_/WO_3_ sample. The WO_3_ and SnO_2_ provide preferential nucleation sites for the following component and preserve a close interface contact between the components [41].

EDX analysis was used to evaluate the elemental composition of the samples and to calculate the component ratio. The analyses were done in three different areas of the samples and the results were similar (<2% abatement), which is an indicator of sample-component homogeneous distribution. The results were compared with the theoretical values calculated based on the stoichiometric compounds, and the results are presented in Table 2. In all three samples, oxygen excess was identified due to the WO_3_ and SnO_2_/WO_3_ thermal treatment at 410 °C in oxygen-rich atmosphere. However, the heterostructure oxygen excess decreases due to the Cu_2_O addition, which was not subjected to annealing [42,43]. During the annealing treatment, the number of oxygen vacancies is diminished due to the oxygen diffusion. The copper ratio compared with the other metal ions is similar which confirms that the weight–ratio relationship between the components was maintained during the synthesis and film deposition.

### 3.2. Photocatalytic Activity

Three pesticide molecules were chosen based on their toxicity and residual persistence in water and soil. Pirimicarb (PIR) is an insecticide that can cause skin irritation, damage to brain functions, and is suspected of causing cancer [44,45]. S-metolachlor (S-MCh) is a herbicide inducing allergic skin irritation, anemia, and has a very toxic potential to aquatic life with long-lasting effects [46,47]. Metalaxyl (MET) is a fungicide with direct impact on the respiratory system and has long persistence in the aquatic environment [48]. This study integrated the large spectrum of pesticide to compare the influence of the photocatalysts on the pollutant molecule. It must be underlined that all the experimental conditions in terms of pollutant concentration, photocatalyst dosage, radiation type, and exposure periods were the same.

#### 3.2.1. Degradation Efficiency and Kinetics

The lowest photocatalytic efficiencies (Figure 4a,c,e) correspond to bare Cu_2_O sample, which exhibit 10.5%, 11.7%, and 8.2% toward PIR, S-MCh, and MET, respectively. Similar photocatalytic activities were obtained for SnO_2_ and Cu_2_O/WO_3_, where coupling Cu_2_O with WO_3_ induced the presence of a high potential gap between the valence energy of Cu_2_O and that corresponding to WO_3_. The WO_3_ monocomponent sample was able to remove 29.8% of PIR, 32.5% of S-MCh, and 12% of MET. Due to this band-gap energy (3.31 eV), the WO_3_ photocatalyst used only a small fraction of the light radiation found at the border between UV and Vis spectra. When the second component is added, the coupled SnO_2_/WO_3_ benefits from the synergic effect of simultaneous charge carrier photogeneration in both semiconductors [49]. The photocatalytic removal efficiencies corresponding to SnO_2_/WO_3_ sample increase at 41.5% for PIR, 53.1% for S-MCh, and 26.8% for MET. The higher photocatalytic efficiencies correspond to the Cu_2_O/SnO_2_/WO_3_ heterostructure, which uses the extended UV and Vis spectra due to the Cu_2_O insertion with a band gap of 2.14 eV. In this case, the photocatalytic removal efficiency was 79.7% for PIR, 86.5% for S-MCh, and 50.6% for MET. The heterostructure photocatalytic removal efficiency is not a sum of the monocomponent sample efficiencies, as the contact between components induces a shift of the energy bands with consequences on the charge carrier’s mobility. The lower MET photocatalytic removal efficiency values compared with the other pesticides indicate that the oxidation process is influenced by the pollutant molecule structure and the chemistry compatibility with the photocatalyst surface [50].

The kinetic evaluation was done using the simplified Langmuir–Hinshelwood mathematical Equation (3) and the results are presented in Figure 4b,d,f.
(3)$$\ln\frac{C}{C_{0}}=-kt$$

The results indicate superior constant rates for S-MCh removal compared with PIR and MET pollutants. Additionally, for all three pollutants the photocatalytic activity of the Cu_2_O/SnO_2_/WO_3_ heterostructure is 2× faster than that of SnO_2_/WO_3_ and 3× faster than that of the WO_3_ monocomponent sample. As predicted, the lowest constant rates correspond to bare Cu_2_O, which exhibits limited photocatalytic activity. However, when Cu_2_O is put in contact with SnO_2_ the constant rates increase 3×, which is a clear indicator of the heterostructure’s ability to decrease the recombination rates and efficiently use the charge carries for oxidative species generation. These values indicate the contribution of each heterostructure component on the overall photocatalytic efficiency toward the pollutant’s molecules. The heterostructure synthesis method has allowed the formation of stable interfaces that maintain high photocatalytic activity during the light irradiation [51].

In order to differentiate the mineralization induced by the photogenerated oxidative species from the partial oxidation products, TOC and TN were investigated (Figure 5a,c,e), as well as the corresponding kinetics (Figure 5b,d,f and Table 3). The TOC measurements for PIR and S-MCh indicate small differences on the photocatalytic activity. The lowest TOC reduction was registered for bare Cu_2_O, followed by Cu_2_O/SnO_2_ and bare SnO_2_. The TOC reduction corresponding to the WO_3_ sample was 27.3% for PIR and 27.9% for S-MCh. Similarly, SnO_2_/WO_3_ (39.3% for PIR and 46.7% for S-MCh) and Cu_2_O/SnO_2_/WO_3_ (73.4% for PIR and 81.3% for S-MCh) recorded slightly TOC reduction efficiency, indicating that not all the carbon components were mineralized. However, the TN evaluation confirms the photocatalytic activity presented in Figure 4, meaning that the nitrogen removed during the irradiation was completely mineralized. An interesting finding was observed for MET pollutant, where both TOC and TN confirm that the photocatalytic removal efficiencies can be attributed to mineralization. It must be outlined that more work must be done to increase the photocatalytic efficiency above 86% for S-MCh, 80% for PIR, and 50% for MET in order to have a significant impact on the long-term effects. The mineralization mechanism of pesticides considers the following steps: (i) charge carriers’ development during the irradiation (Equation (4)); (ii) oxidative species (HO·) production during the reaction between the photogenerated holes and water molecules (Equation (5)); (iii) superoxidative species (·O_2_^–^) development during the reaction between photogenerated electrons and dissolved oxygen (Equation (6)), and (iv) interaction between the (super)oxidative species and pesticide molecules (Equation (7)).
Heterostructure + h*ν* → e^−^ + h^+^(4)
h^+^ (Heterostructure) + H_2_O → HO⋅ + H^+^(5)
O_2_ + e^−^ (Heterostructure) → ⋅O^−^_2_(6)
Pesticides + HO⋅ + ⋅O^−^_2_ → xCO_2_ + yH_2_O + NO_2_(7)

The kinetic evaluation indicates that the WO_3_ monocomponent sample has small and similar constant rates for all three pollutant molecules, which is a consequence of reduced light absorbance and high charge carrier recombination. Larger differences in the constant rates were obtained for SnO_2_/WO_3_ and Cu_2_O/SnO_2_/WO_3_ samples, which confirm the photocatalytic removal efficiency results. The insertion of an additional semiconductor component will increase the photocatalytic efficiency where each partner will play a significant role on the charge carrier photogeneration and mobility. These characteristics are essential in order to produce oxidative (·OH) and superoxidative (·O_2_^−^) species able to induce pollutant mineralization [52,53,54].

#### 3.2.2. Reusability and Mechanism of Charge Carrier Generation

The mechanism of charge carrier generation through the heterostructure can explain the role of each component in enhancing the photocatalytic efficiency toward the pesticide molecules. The band energy diagram (Figure 6a) was constructed considering the experimental band gap corresponding to each heterostructure component based on the Wood and Tauc model, as presented in Figure 6b–d. The methodology has been already presented in another paper [55] and is in good agreement with the literature [56,57]. The band gap may submit minor shift during the heterostructure development. The methodology takes into consideration the band-gap changes during the heterostructure’s internal energy field developed during the irradiation. The diagram includes the energy band position based on Equations (8)–(11), which consider several key parameters: E_e_, representing the free electron energy vs. hydrogen, χ_cation_ (eV), representing the absolute cationic electronegativity, χ_cation_ (P.u.) is the cationic specific electronegativity, where P.u. corresponds to the Pauling units, E_g_ is the band-gap energy, and χ_semiconductor_ represents the electronegativity of each semiconductor.
E_VB_ = χ_semiconductor_ − E_e_ + 0.5E_g_(8)
E_CB_ = E_VB_ − E_g_(9)
χ_semiconductor_ (eV) = 0.45 × χ_cation_ (eV) + 3.36(10)
(11)$$\chi_{\text{cation}}(eV)=\frac{\chi_{\text{cation (P.u.)}}+0.206}{0.336}$$


During the semiconductors’ interface development, the band gap may shift due to the internal energy field. The diagram corresponds to a Z-scheme charge transfer mechanism where the electrons generated during the light irradiation from the Cu_2_O conduction band (−0.35 eV) will transit the SnO_2_ conduction band on the way to the WO_3_ conduction band (+0.14 eV). The photogenerated electrons from SnO_2_ and WO_3_ conduction bands and the photogenerated holes from the Cu_2_O valence band (+1.79 eV) are not involved in oxidative species development, owing to their potential. In this case, some of the charge carriers will recombine. However, the useful photogenerated electrons from the Cu_2_O conduction band and the photogenerated holes from the SnO_2_ (+2.45 eV) and WO_3_ (3.44 eV) valence bands possess stronger redox ability and cannot recombine due to the electric field development in the charged separation region. The charge carrier’s mobility is sustained by the combined drift and diffusion effect. Consequently, the synergic effect provided by the heterostructure semiconductor components enhances the production of oxidative and superoxidative radicals responsible for pollutant mineralization. The heterostructure requires further improvement in order to reduce the charge recombination and to increase the overall photocatalytic efficiency toward pesticides.

The reusability evaluation (Figure 7) was performed on the samples with the highest photocatalytic efficiency using a three-cycle assessment. The results indicate that for all three pollutant molecules, the Cu_2_O/SnO_2_/WO_3_ exhibits good stability with negligible photocatalytic changes between cycles. The WO_3_ and SnO_2_/WO_3_ show small abatement of less then 5%, mostly on the second cycle. The changes can be induced by pollutant molecule adsorption at the photocatalyst surface active centers, which may require a longer desorption/degradation period. The results indicate that the Cu_2_O/SnO_2_/WO_3_ heterostructure can be used for multiple cycle assessment without influencing the photocatalytic activity toward pesticide molecules.

## 4. Conclusions

A heterostructure based on Cu_2_O/SnO_2_/WO_3_ was synthesized using a three-step sol–gel method and deposited as films on glass substrate by spray deposition. The sample contains tetragonal SnO_2_ and monoclinic WO_3_ with similar crystallite sizes (≈ 80 Å), while cubic Cu_2_O exhibits significantly lower crystallite sizes (≈ 30 Å). The porous morphology of the final heterostructure was constructed starting with fiber-like WO_3_, which serves as substrate for SnO_2_ development, and SnO_2_/WO_3_ provides the nucleation and growing sites for Cu_2_O formation. The elemental analysis confirms that the samples are homogeneous in composition and exhibit oxygen excess due to the annealing treatments. The EDX measurements were done on three different areas, indicating that the final heterostructure preserves the component:weight ratio used during the synthesis.

Three pesticide molecules (pirimicarb, S-metolachlor, and metalaxyl) were used as pollutants reference and were chosen based on their resilience and toxicity. The photocatalytic activity of the Cu_2_O/SnO_2_/WO_3_ heterostructure is superior to that of bare oxides or tandem systems. The lowest photocatalytic activity (≈ 10%) corresponds to bare Cu_2_O, followed by SnO_2_ and Cu_2_O/WO_3_. The highest photocatalytic efficiency was obtained toward S-MCh (86%) using the Cu_2_O/SnO_2_/WO_3_ sample and the lowest corresponds to MET (12%) removal using the WO_3_ monocomponent sample. However, the TOC analysis indicates lower mineralization efficiencies that can be attributed to secondary product formation. The photocatalytic mechanism corresponds to a Z-scheme charge transfer where Cu_2_O exhibits reduction potential and WO_3_ oxidation potential. The charge carrier’s mobility sustained by the combined drift and diffusion effect provides a synergic effect where the heterostructure semiconductor components enhance the (super)oxidative radicals’ production. The reusability tests indicate that Cu_2_O/SnO_2_/WO_3_ exhibits similar photocatalytic efficiency after a three-cycle assessment, which recommends this material for further experiments.

## Figures and Tables

**Figure 1 nanomaterials-12-02648-f001:**
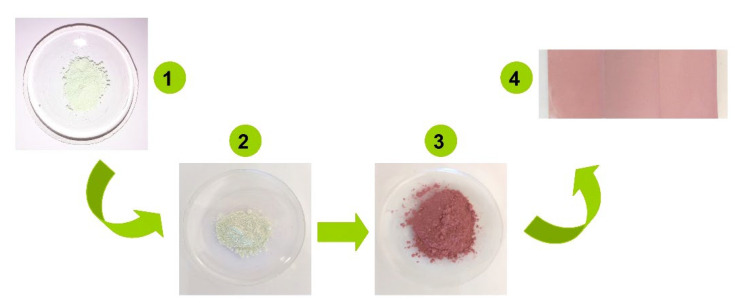
Steps for heterostructure development.

**Figure 2 nanomaterials-12-02648-f002:**
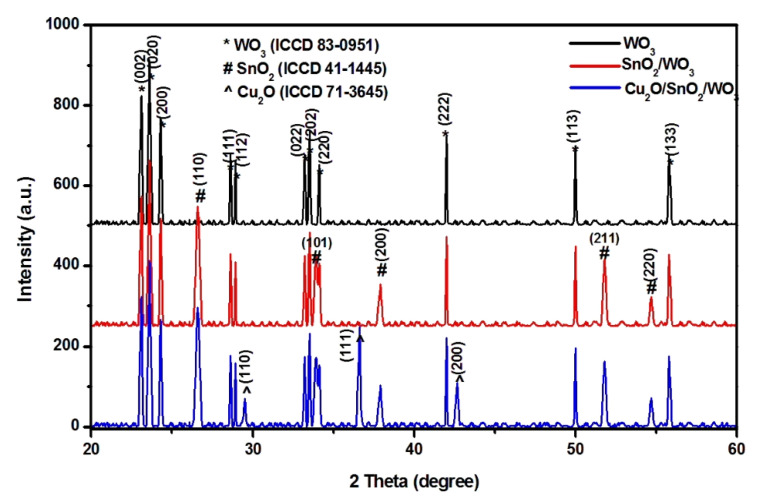
XRD patterns of the photocatalysts.

**Figure 3 nanomaterials-12-02648-f003:**
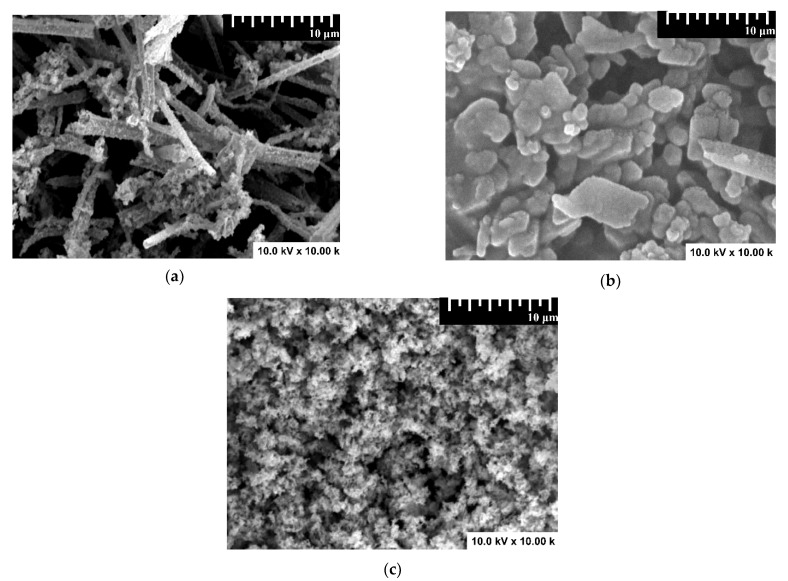
SEM images of (**a**) WO_3_, (**b**) SnO_2_/WO_3_, and (**c**) Cu_2_O/SnO_2_/WO_3_ samples.

**Figure 4 nanomaterials-12-02648-f004:**
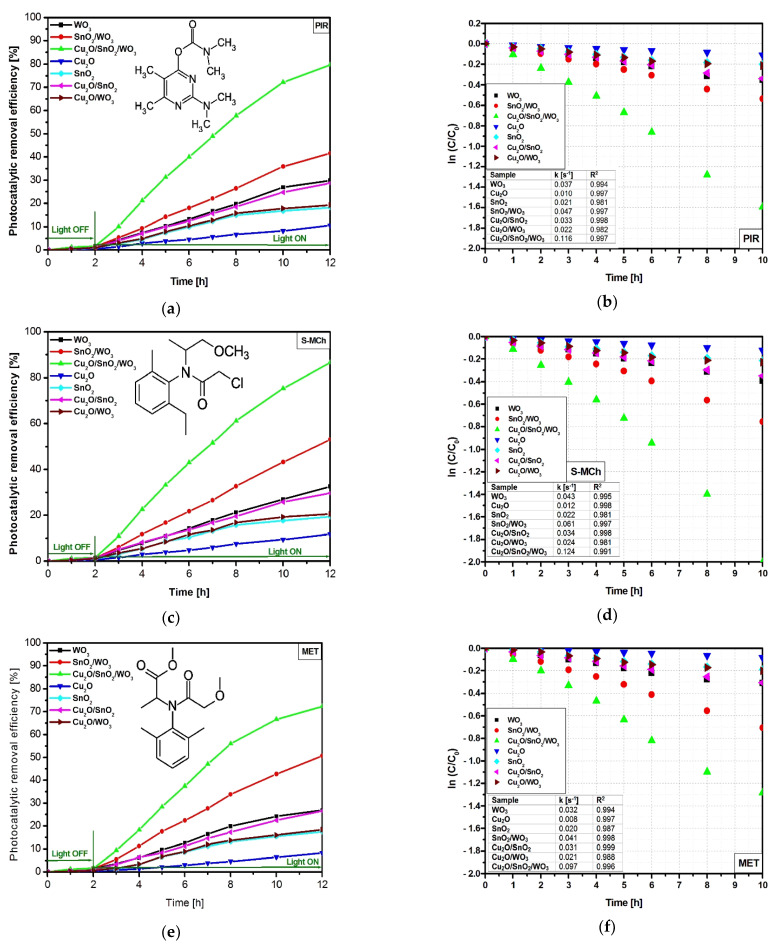
Photocatalytic removal efficiency and the corresponding kinetics toward PIR (**a**,**b**), S-MCh (**c**,**d**), and MET (**e**,**f**).

**Figure 5 nanomaterials-12-02648-f005:**
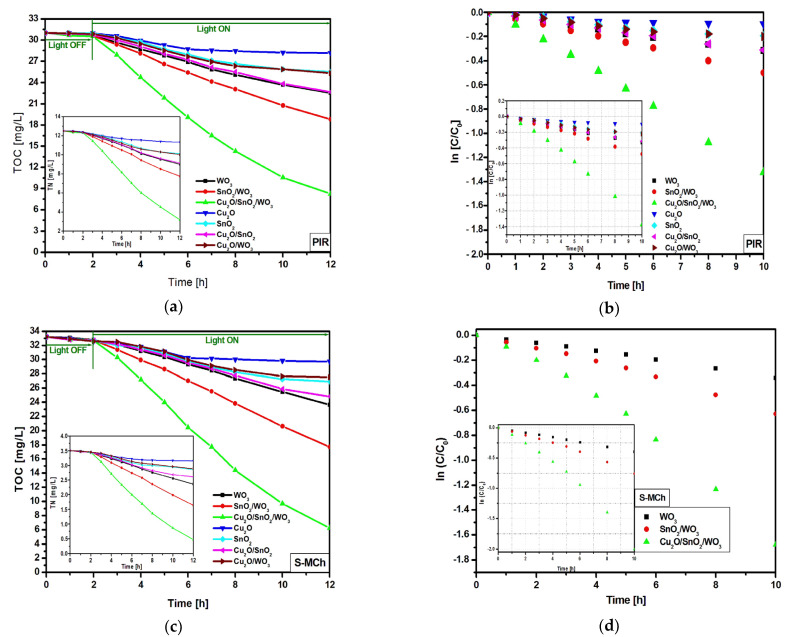
TOC, TN (inset), and the corresponding kinetics toward PIR (**a**,**b**), S-MCh (**c**,**d**), and MET (**e**,**f**).

**Figure 6 nanomaterials-12-02648-f006:**
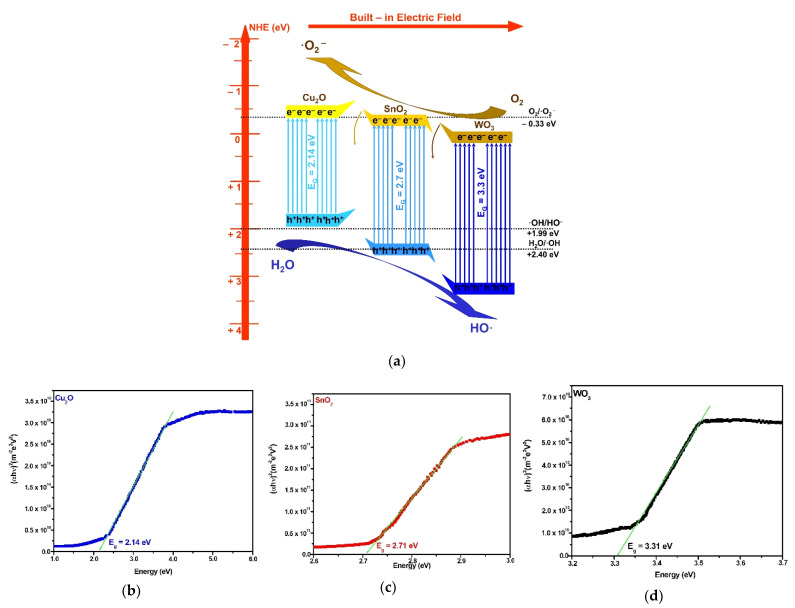
Heterostructure mechanism (**a**) and component band-gap values (**b**–**d**).

**Figure 7 nanomaterials-12-02648-f007:**
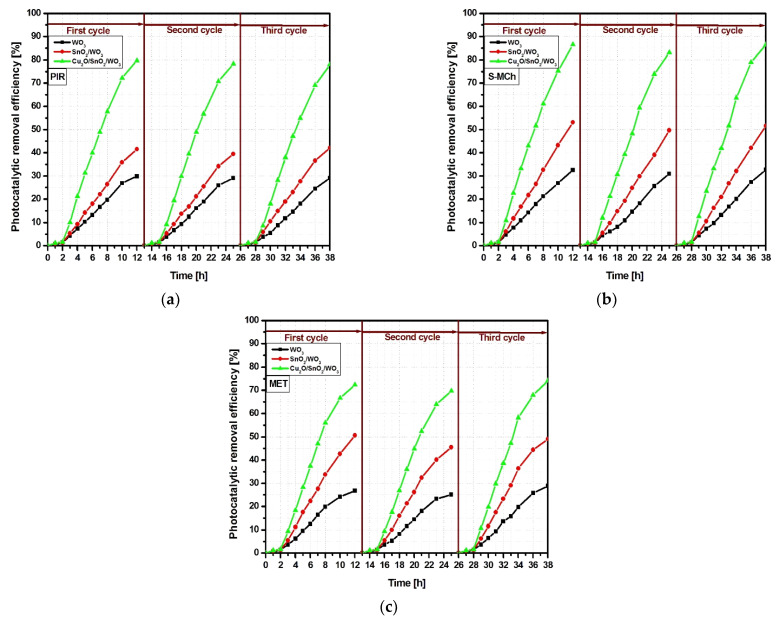
Photocatalytic activity during the 3-cycle sample reusability for (**a**) PIR, (**b**) S-MCh and (**c**) MET pesticides.

**Table 1 nanomaterials-12-02648-t001:** Photocatalyst crystallite size values evaluated based on Scherrer formula.

Photocatalyst	Crystallite Size (Å)		
	WO_3_	SnO_2_	Cu_2_O
WO_3_	80.5	-	-
SnO_2_/WO_3_	81.8	77.1	-
Cu_2_O/SnO_2_/WO_3_	81.6	77.3	42.4

**Table 2 nanomaterials-12-02648-t002:** Photocatalyst elemental composition by EDX.

Samples	Elemental Composition [at.%]				
	W	Sn	Cu	O	O_th_ ^1^
WO_3_	23.8	-	-	76.2	71.4
SnO_2_/WO_3_	12.4	14.7	-	72.9	68.9
Cu_2_O/SnO_2_/WO_3_	10.6	11.9	13.3	64.2	62.2

^1^ Theoretic content calculated based on stoichiometry.

**Table 3 nanomaterials-12-02648-t003:** Kinetic parameters for TOC and TN evaluation.

Kinetic Parameters		PIR		S-MCh		MET	
		TOC	TN	TOC	TN	TOC	TN
**WO_3_**	k (s^−1^)	0.031	0.033	0.033	0.038	0.030	0.032
	R^2^	0.997	0.998	0.998	0.999	0.998	0.994
Cu_2_O	k (s^−1^)	0.010	0.009	0.011	0.009	0.006	0.005
	R^2^	0.911	0.947	0.919	0.928	0.920	0.866
SnO_2_	k (s^−1^)	0.020	0.022	0.022	0.019	0.019	0.020
	R^2^	0.978	0.989	0.982	0.965	0.981	0.969
SnO_2_/WO_3_	k (s^−1^)	0.049	0.047	0.062	0.074	0.045	0.043
	R^2^	0.999	0.998	0.995	0.994	0.998	0.998
Cu_2_O/SnO_2_	k (s^−1^)	0.031	0.032	0.030	0.030	0.030	0.031
	R^2^	0.998	0.998	0.998	0.989	0.999	0.999
Cu_2_O/WO_3_	k (s^−1^)	0.021	0.022	0.020	0.019	0.020	0.020
	R^2^	0.972	0.987	0.975	0.978	0.973	0.963
Cu_2_O/SnO_2_/WO_3_	k (s^−1^)	0.135	0.137	0.168	0.195	0.120	0.119
	R^2^	0.998	0.994	0.990	0.986	0.997	0.998

## Data Availability

Data presented in this study are available upon request from the corresponding author.

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
