# Peer review of "UV-Vis Activated Cu2O/SnO2/WO3 Heterostructure for Photocatalytic Removal of Pesticides"

_nanomaterials, 2022, doi:10.3390/nano12152648_

Round 1

Reviewer 1 Report

This manuscript focuses on Sol-gel methods in order to obtain Cu2O/SnO2/WO3 heterostructure powder, deposed as film by spray pyrolysis.

The photocatalytic properties of the as-grown films were tested in the presence of three pesticides (pirimicarb PIR, S-metolachlor S-MCh and metalaxyl MET) chosen based on their resilience and toxicity.

As the authors state, the photocatalytic activity of the Cu2O/SnO2/WO3 heterostructures is superior to that of SnO2/WO3 and WO3 samples.

This is an interesting work; Nevertheless some minor revisions are needed in order to publish this work.

1. As the authors state "..The light sources are placed in radial position to provide the maximum light radiation and to ensure the uniform light distribution on the samples.." What is the energy of the incident light on the samples (in mW/cm2). This information is essential in order to compare these findings with the literature.

2. Uv-vis spectroscopy cannot identify "photocatalytic activity". You can only detect decolorization. After total organic carbon (TOC) measurements the authors can discuss about the quantity of pollutant completely mineralized during the photocatalytic reaction.  I kindly ask the authors to rephrase the UV-Vis analysis.

3. Can the author comment on the re-use of their samples after 3-5 times? the re-usability is a great parameter that should be taken into account.

Author Response

Dear Reviewer,

We express our gratitude to your work and guidance that was helping us to improve the quality of this manuscript.

We have considered all your comments and suggestion in the new revised form of the manuscript. The changes where highlighted in red.

This is an interesting work; Nevertheless some minor revisions are needed in order to publish this work.

Q1. As the authors state "..The light sources are placed in radial position to provide the maximum light radiation and to ensure the uniform light distribution on the samples.." What is the energy of the incident light on the samples (in mW/cm2). This information is essential in order to compare these findings with the literature.

A1. Thank you for the suggestion. We have inserted the energy of the incident light.

Lines: 124-126

Other changes made to the Experimental part were done to make this subchapter clearer and easier to follow.

Lines: 87-89

            94-96

            100-104

108-110

Q2. Uv-vis spectroscopy cannot identify "photocatalytic activity". You can only detect decolorization. After total organic carbon (TOC) measurements the authors can discuss about the quantity of pollutant completely mineralized during the photocatalytic reaction.  I kindly ask the authors to rephrase the UV-Vis analysis.

A2. Thank you for the suggestion. We made the changes accordingly.

Lines: 151-156

Q3. Can the author comment on the re-use of their samples after 3-5 times? the re-usability is a great parameter that should be taken into account.

A3. Thank you for the suggestion. The 3 cycle’s re-usability tests were inserted.

Lines: 314-321

Figure 7a-c

Reviewer 2 Report

The current manuscript entitled “UV-Vis activated Cu2O/SnO2/WO3 heterostructure for photocatalytic removals of pesticides synthesized Cu2O/SnO2/WO3 heterostructure using a three-step sol-gel method and was deposed as films by spray deposition. The photocatalytic activity of the Cu2O/SnO2/WO3 heterostructure is superior to that of SnO2/WO3 and WO3 samples. The subject is particularly important from a practical standpoint, however, there is no evidence for the heterojunction formation. The authors made a statement without proper catalyst characterization/pieces of evidence. I don’t recommend the manuscript in its current form for publication.  

The detailed revision are as follows   

  1. The authors should include the motivation behind choosing this Cu2O/SnO2/WO3 heterostructure in the introduction part. 
  2. On page no: 2 and Line no: 79 and 87, Please correct the Natium to Sodium hydroxide. 
  3. Please correct the mistakes in the manuscript title. UV-Vis activated Cu2O/SnO2/WO3 heteorstructure for photocatalitic removal of pesticides. 
  4. Figures representation needs to be improved. 
  5. What is the status of Cu Oxidations state? Why there is no calcination after Cu deposition? 
  6. On page no. 6 and line no. 179 the authors stated, “The final heterostructure containing all three components presents a porous morphology (Figure 3c) with Cu2O uniformly distributed on the sample surface.” However, microscopic image quality is poor and cannot show any uniform elemental distribution. 
  7. The manuscript's English needs to be improved. 
  8. The obtained catalytic results just provide an insight into the conversion performance of as-prepared samples. No accurate conclusions were obtained about the intrinsic activity of the samples. 
  9. Although Cu2O/SnO2/WO3 activity results are more compared to its counterparts, more shreds on the catalyst characterization are required to know the role of Cu2O/SnO2/WO3. 

Author Response

Dear Reviewer,

We express our gratitude to your work and guidance that was helping us to improve the quality of this manuscript.

We have considered all your comments and suggestion in the new revised form of the manuscript. The changes where highlighted in red.

The detailed revision are as follows   

 Q1. The authors should include the motivation behind choosing this Cu2O/SnO2/WO3 heterostructure in the introduction part. 

A1. We appreciate your sugestion. The Introduction part was improved accordingly.

Lines: 54-59

            66-72

Q2. On page no: 2 and Line no: 79 and 87, Please correct the Natium to Sodium hydroxide. 

A2. We made the corrections.

Lines: 88 and 96

Q3. Please correct the mistakes in the manuscript title. UV-Vis activated Cu2O/SnO2/WO3 heteorstructure for photocatalitic removal of pesticides. 

A3. Thank you for the observation. We made the correction.

Line: 2

Q4. Figures representation needs to be improved. 

A4. We have improved the figures resolution.

Q5. What is the status of Cu Oxidations state? Why there is no calcination after Cu deposition? 

A5. The cooper oxidation state is +1. We have avoided the calcination process due to the cooper tendency to change in +2 oxidation state very easly. Even now, we can’t exclude the presence of CuO in very small amouth or in amorphous state.

Lines: 100-104

Q6. On page no. 6 and line no. 179 the authors stated, “The final heterostructure containing all three components presents a porous morphology (Figure 3c) with Cu2O uniformly distributed on the sample surface.” However, microscopic image quality is poor and cannot show any uniform elemental distribution. 

A6. Thank you for the comment. We have improved the resolution figure as much as possible. However, due to the poor surface conductivity the SEM images are not as clear as desired. The uniform Cu2O distribution was confirmed by EDS analysis made on multiple areas with similar results (an abatment of < 2% was observed).

Q7. The manuscript's English needs to be improved. 

A7. The manuscript English was revised accordingly.

Q8. The obtained catalytic results just provide an insight into the conversion performance of as-prepared samples. No accurate conclusions were obtained about the intrinsic activity of the samples. 

Q9. Although Cu2O/SnO2/WO3 activity results are more compared to its counterparts, more shreds on the catalyst characterization are required to know the role of Cu2O/SnO2/WO3. 

A8 & A9. Thank you for the comments. We have considered your observation and extensive changes were done to the photocatalytic evaluation. Four new samples were inserted. All graphs were changed to include the new samples. New experiments were done to evaluate the re-usability. Table 3 was upgraded to include all samples.

Lines: 230-234

            243-245

            254-257

            267-272

            314-321

Table 3

Figures 4, 5 and 7

Reviewer 3 Report

Review of the nanomaterials-1777889 for the Authors: This work investigates UV-Vis activated oxide heterostructures for photocatalytic removal of pesticides. The results are mainly on (micro)structural properties of Cu2O/SnO2/WO3 films in combination with elemental and photocatalytic study. The photocatalytic activity at three pesticides is demonstrated as superior for Cu2O/SnO2/WO3 in comparison to SnO2/WO3 and WO3. The authors leave us with general suggestion for further optimization of the samples to increase their photocatalytic impact and possible application. While the results presented seem to be interesting, they are generally not interconnected with discussion. Results from different sections should be better correlated and explained to reach the level suitable for publication.

Check title spelling.

Abstract and Conclusions: In both, it seems that more connections and explanations should be given with more obtained results. These sections are too similar.

Introduction - more about the objective of the present investigation. Why these samples are interesting to explore? What about similar systems? Reduce number of references.

Experimental, Characterization - more details are necessary.

Results and discussion: PXRD: Diffraction lines? Some lines are out of the frame. Did you notice preferred orientation? Comment. Are the characterization methods done on powder or films?

Experimental: In general, please make it easier to follow.

Results and discussion: SEM. Please comment on crystallite size determined from XRD and particle size obtained from SEM?

EDS. Last sample. Please Comment on amount of Cu and its ratio in comparison to Sn and W.

Photocatalytic study. Please comment more into details the comparison of obtained results on type of pesticide (PIR, S-McH and MET,).

Mechanism 3.2.2. It would be good to incorporate this paragraph in through whole results a discussion section, and not put it at the end of the paper.

Please check the language style and grammar through whole manuscript.

I recommend major revision.

Author Response

Dear Reviewer,

We express our gratitude to your work and guidance that was helping us to improve the quality of this manuscript.

We have considered all your comments and suggestion in the new revised form of the manuscript. The changes where highlighted in red.

Q1. Abstract and Conclusions: In both, it seems that more connections and explanations should be given with more obtained results. These sections are too similar.

A1. Thank you for the suggestions. We made the revision accordingly.

Lines: 16-22

            331-333

            335-336

            344-346

Q2. Introduction - more about the objective of the present investigation. Why these samples are interesting to explore? What about similar systems? Reduce number of references.

A2. We have improved the Introduction part by adding supplementary explanations and information’s.

Lines: 54-59

            66-72

Q3. Experimental, Characterization - more details are necessary.

A3. Thank you for the suggestion. The experimental and characterization part was improved in order to be clearer and easier to follow.

Lines: 87-89

            94-96

            100-104

            108-110

            124-126

            151-156

Q4. Results and discussion: PXRD: Diffraction lines? Some lines are out of the frame. Did you notice preferred orientation? Comment. Are the characterization methods done on powder or films?

A4. The diffraction line was inserted. After results superposition the frame was enlarge to include all the lines. All the characterization were done on films which were the also used for photocatalytic experiments. We didn’t notice a preferred orientation maybe to the synthesis method used in this case.

Lines: 163-164

Figure 2

Q5. Experimental: In general, please make it easier to follow.

A5. Thank you for the suggestion. The experimental and characterization part was improved in order to be clearer and easier to follow.

Lines: 87-89

            94-96

            100-104

            108-110

            124-126

            151-156

Q6. Results and discussion: SEM. Please comment on crystallite size determined from XRD and particle size obtained from SEM?

A6. Thank you for the suggestion. We have added new comments in order to give more explanation related with the crystallite size. The particle size was not evaluated considering that WO3 is fiber-like, and the heterostructure morphology doesn’t allow a clear distinction between particles.

Lines: 172-173

            182-184

            193-194

            198-200

Q7. EDS. Last sample. Please Comment on amount of Cu and its ratio in comparison to Sn and W.

A7. We have added new comments related with the Cu ratio.

Lines: 204-206

            210-213

Q8. Photocatalytic study. Please comment more into details the comparison of obtained results on type of pesticide (PIR, S-McH and MET,).

Q9. Mechanism 3.2.2. It would be good to incorporate this paragraph in through whole results a discussion section, and not put it at the end of the paper.

A8 and A9. Thank you for the comments. We have considered your observation and extensive changes were done to the photocatalytic evaluation. Four new samples were inserted. All graphs were changed to include the new samples. New experiments were done to evaluate the re-usability. Table 3 was upgraded to include all samples.

Lines: 230-234

            243-245

            254-257

            267-272

            314-321

Table 3

Figures 4, 5 and 7

Q10. Please check the language style and grammar through whole manuscript.

A10. The manuscript was check for language and grammar errors.

Round 2

Reviewer 1 Report

The authors have improved their manuscript following the suggested revisions.

This work can be published in its present form.

Author Response

Thank you!

Reviewer 2 Report

Although, current manuscript entitled UV-Vis activated Cu2O/SnO2/WO3 heterostructure for photocatalytic removal of pesticides, claiming the heterostructure formation, authors did not show any experimental evidence. The XRD and SEM images qualities are very poor. The current version of the manuscript is not suitable for Nanomaterials.  
  1. Please correct the mistakes in the manuscript title. UV-Vis activated Cu2O/SnO2/WO3 heterostructure for photocatalitic removal of pesticides. 
  1. Figures representation must be improved. 
  1. Microscopic image quality is poor and cannot show any uniform elemental distribution.
  2.  
  3. The obtained catalytic results just provide an insight into the conversion performance of as-prepared samples. No accurate conclusions were obtained about the intrinsic activity of the samples.
  4.  
  5. Although Cu2O/SnO2/WO3 activity results are more compared to its counterparts, more shreds on the catalyst characterization are required to know the role of Cu2O/SnO2/WO3. 

Author Response

Dear Reviewer,

We express our gratitude to your work and guidance that was helping us to improve the quality of this manuscript.

We have considered all your comments and suggestion in the new revised form of the manuscript. The changes where highlighted in red.

Q1. Please correct the mistakes in the manuscript title. UV-Vis activated Cu2O/SnO2/WO3 heterostructure for photocatalitic removal of pesticides. 

A1. Thank you for the remark. We made the secondarry correction.

Lines: 2-3

Abstract, Introduction and Conclusions part were improved:

Lines: 12-14

            22-29

            75-77

            351

357-362

364-372

Q2. Figures representation must be improved. 

A2. We done our best effort to improve the figures representation. All Figures have the standard resolution that is accepted in Nanomaterials.

Q3. Microscopic image quality is poor and cannot show any uniform elemental distribution.

A3. We understand your concerns about the SEM images quality. We are restricted by the available research infrastructure technology and samples characteristics. We have made several attempt to increase the magnitude but we lose considerable resolution. Metal covarage was also attempted but in this case elemental distribution evaluation is not longer possible.

Q4. The obtained catalytic results just provide an insight into the conversion performance of as-prepared samples. No accurate conclusions were obtained about the intrinsic activity of the samples.

A4. The photocatalytic evaluation was significantly improved with more details and explanations.

Equations: 4, 5, 6, 7, 8, 9, 10, 11.

Lines: 285-289

            311-318

Q5. Although Cu2O/SnO2/WO3 activity results are more compared to its counterparts, more shreds on the catalyst characterization are required to know the role of Cu2O/SnO2/WO3. 

A5. The photocatalytic evaluation was significantly improved with more details and explanations.

Equations: 4, 5, 6, 7, 8, 9, 10, 11.

Lines: 285-289

            311-318

Additionaly, more explanations were inserted in the Results and Discussion chapter.

Lines: 168-169

            175-176

            178-179

            185-187

            200-201

            211-213

Reviewer 3 Report

Review of the nanomaterials-1777889-r2 for the Authors: I would like to thank Authors for taking into account suggestions mentioned and improved the quality of manuscript making changes in the text and answering questions. The effort is really appreciated, however, in my opinion, some parts are still not clear and commented. Please see below. Thank you in advance.

Abstract and Conclusions: The connection part is still not covered, please go through whole parts, and make it in whole more connected and easier to follow.

Introduction: Please reduce the number of references; but too many information just without connection; a bit more why those samples are important and interesting.

Experimental, Characterization: You prolonged the Experimental part, but Characterization is still poor in terms of details such as wavelength, range, etc.

Results and discussion: Include answers into the text.

Photocatalytic study could still benefit from upgrading the mechanism paragraph. Please check English!

I recommend major revision until this is settled.

Author Response

Dear Reviewer,

We express our gratitude to your work and guidance that was helping us to improve the quality of this manuscript.

We have considered all your comments and suggestion in the new revised form of the manuscript. The changes where highlighted in red.

Q1. Abstract and Conclusions: The connection part is still not covered, please go through whole parts, and make it in whole more connected and easier to follow.

A1. Thank you for the observation. We improved both Abstract and Conclusions to be easier to follow.

Lines: 12-14

            22-29

            351

357-362

364-372

Q2. Introduction: Please reduce the number of references; but too many information just without connection; a bit more why those samples are important and interesting.

A2. We have reduced the number of references (8 references were removed: 7 from Introduction and 1 from Dicussion). The informations were correlated starting form mono-component catayst to tandem systems. A new sentance was inserted related to the materials used in this study.

Lines: 75-77

Q3. Experimental, Characterization: You prolonged the Experimental part, but Characterization is still poor in terms of details such as wavelength, range, etc.

A3. Thank you for the comment. We include more details on the Characterization part. In this sense we include the photocatalysis evaluation in Characterization section. Experimental is always longer due to the synthesis details and photocatalytic conditions.

Lines: 145

            154-163

Q4. Results and discussion: Include answers into the text.

A4. We have included all the answers into the text.

Equations: 4, 5, 6, 7, 8, 9, 10, 11.

Lines: 168-169

            175-176

            178-179

            185-187

            200-201

            211-213

            285-289

            311-318

Q5. Photocatalytic study could still benefit from upgrading the mechanism paragraph.

A5. We made significant improvement on explaining the intrinsic and extrinsic mechanisms.

Equations: 4, 5, 6, 7, 8, 9, 10, 11.

Lines: 285-289

            311-318

Q6. Please check English!

A6. The English was verified. Small mistakes not detected at this stage will be corrected during the manuscript processing.

Round 3

Reviewer 2 Report

The authors’ response and revisions have satisfactorily addressed my comments on the earlier version of the manuscript.  I support publication of this version and look forward to reading the follow-up studies that utilize this work.

Author Response

Thank you!

Reviewer 3 Report

Review of the Nanomaterials-1777889-r3 for the Authors:

Dear Authors, thank you very much for taking into account suggestion proposed and changes which are done accordingly. Improvement is visible. However, I would like you to comment few more points below:

1.       Page2, Line 69–82: All sentences start with The

2.       Experimental, P3, Line 119: Films are sprayed, how? Please share more details.

3.       P3, Line 144: step is 0.002 or 0.02?

4.       P4 Line 167–179: Again, article The

5.       XRD, crystallite size, page 5: Did you take into account instrumental broadening? Not mentioned in the text.

6.       EDS. Last sample. Please Comment on amount of Cu and its ratio in comparison to Sn and W. Why it seems Cu is lower more than 2at%

7.       Please check English and spelling

I recommend major revision.

Author Response

Dear Reviewer,

We express our gratitude to your work and guidance that was helping us to improve the quality of this manuscript.

We have considered all your comments and suggestion in the new revised form of the manuscript. The changes where highlighted in red.

Dear Authors, thank you very much for taking into account suggestion proposed and changes which are done accordingly. Improvement is visible. However, I would like you to comment few more points below:

Q1.       Page2, Line 69–82: All sentences start with The

A1. Thank you for the suggestion. We have removed and reformulate several sentences to solve these aspects.

Lines: 67-82

  1. Experimental, P3, Line 119: Films are sprayed, how? Please share more details.

A2. Details about the spraying procedure were inserted.

Lines: 119-121

  1. P3, Line 144: step is 0.002 or 0.02?

A3. The step in 0.002 and the scanning time for each step was 0.02s.

  1. P4 Line 167–179: Again, article The

A4. Thank you for the suggestion. We have removed and reformulate several sentences to solve these aspects.

Lines: 167-179

  1. XRD, crystallite size, page 5: Did you take into account instrumental broadening? Not mentioned in the text.

A5. Thank you for the question. Of course, we have considered the instrumental broadening in our calculations. This plays an important role mostly at low intensity diffraction lines. We have included a sentence into the manuscript to make this aspect clearer.

Line: 186

  1. EDS. Last sample. Please Comment on amount of Cu and its ratio in comparison to Sn and W. Why it seems Cu is lower more than 2at%

A6. Based on our evaluation the Cu content is similar with the other metals.

  1. Please check English and spelling

A7. We made a supplementary check on English and spelling.
